# Evaluating the Influence of Mood and Stress on Glycemic Variability in People with T1DM Using Glucose Monitoring Sensors and Pools

Jose Manuel Velasco [1], Marta Botella-Serrano [2], Almudena Sánchez-Sánchez [3], Aranzazu Aramendi [2], Remedios Martínez [2], Esther Maqueda [4], Oscar Garnica [1], Sergio Contador [1], Juan Lanchares [1] and José Ignacio Hidalgo [1,*]

[1] Adaptive and Bioinspired Systems Research Group, Computer Architecture and Automation Department, Universidad Complutense de Madrid, 28040 Madrid, Spain; mvelascc@ucm.es (J.M.V.); ogarnica@ucm.es (O.G.); scontador@ucm.es (S.C.); julandan@ucm.es (J.L.)

[2] Endocrinology and Nutrition Service, Hospital Universitario Príncipe de Asturias, 28805 Alcalá de Henares, Spain; marta.botella@salud.madrid.org (M.B.-S.); arantza.az@hotmail.com (A.A.); maremart-ro@hotmail.com (R.M.)

[3] Health Sciences Faculty, Universidad a Distancia de Madrid, 28400 Madrid, Spain; Almudena.sanchez.s@udima.es

[4] Endocrinology and Nutrition Service, Hospital Virgen de la Salud, 45004 Toledo, Spain; esthemaq@ucm.es

* Correspondence: absys@ucm.es

**Abstract: Objective**: Assess in a sample of people with type 1 diabetes mellitus whether mood and stress influence blood glucose levels and variability. **Material and Methods**: Continuous glucose monitoring was performed on 10 patients with type 1 diabetes mellitus, where interstitial glucose values were recorded every 15 min. A daily survey was conducted through Google Forms, collecting information on mood and stress. The day was divided into six slots of 4-h each, asking the patient to assess each slot in relation to mood (sad, normal or happy) and stress (calm, normal or nervous). Different measures of glycemic control (arithmetic mean and percentage of time below/above the target range) and variability (standard deviation, percentage coefficient of variation, mean amplitude of glycemic excursions and mean of daily differences) were calculated to relate the mood and stress perceived by patients with blood glucose levels and glycemic variability. A hypothesis test was carried out to quantitatively compare the data groups of the different measures using the Student's *t*-test. **Results**: Statistically significant differences (*p*-value < 0.05) were found between different levels of stress. In general, average glucose and variability decrease when the patient is calm. There are statistically significant differences (*p*-value < 0.05) between different levels of mood. Variability increases when the mood changes from sad to happy. However, the patient's average glucose decreases as the mood improves. **Conclusions**: Variations in mood and stress significantly influence blood glucose levels, and glycemic variability in the patients analyzed with type 1 diabetes mellitus. Therefore, they are factors to consider for improving glycemic control. The mean of daily differences does not seem to be a good indicator for variability.

**Keywords:** glucose variability; stress; mood; Continuous Glucose Monitoring; glycemic control

## 1. Background

The prevalence of depression and psycho social stress is higher in people with Type 1 Diabetes Mellitus (T1DM) [1]. The presence of emotional disorders has been associated with poorer long-term glycemic control [2]. The impact of daily mood swings and stress on short-term glycemic control and glycemic variability has been little studied, and to date no study with Continuous Glucose Monitoring (CGM) has been reported. High glycemic variability increases the risk of hypoglycemia, hinders metabolic control, and may be associated with an increase in chronic complications.

Stress is not only related to high blood glucose levels, but there are people in whom stress causes low blood glucose levels [3]. With stress, they lose their appetite and increase their risk of life-threatening hypoglycemia. In some cases, it could lead them to present what is called chronic fatigue [4]. Significant associations between stress and metabolic control have been found in several studies, through the HbA1c level in patients with diabetes [5–8]. Other studies use the The Diabetes Distress Scale [9], where stress was found to be directly and statistically significantly correlated with HbA1c level [10].

This work aims to study whether mood changes and stress influence blood glucose levels in the short-term, relating mood and stress perceived by patients with blood glucose levels and glycemic variability. The study is performed by analyzing two glycemic control metrics (arithmetic mean and percentage of time below/above the target range) and four glycemic variability metrics (standard deviation, percentage coefficient of variation, mean amplitude of glycemic excursions and mean of daily differences).

The rest of the article is organized as follows. Section 2 describes the data and the measures used for glycemic control and the calculation of variability. The results obtained are shown in Section 3. The conclusions are presented in Section 4.

## 2. Material and Methods

### 2.1. Patients

CGM was performed using FreeStyle Libre sensors from Abbott, on 10 patients with T1DM where interstitial glucose values were recorded every 15 min. FreeStyle Libre sensors were selected because some of the patients were covered by the Spanish National Health System and they already knew the particularities of those sensors. A daily survey was developed through Google Forms, collecting information on mood and stress. The form is available through an online link (esp, https://forms.gle/ehcoo6hGrbKouscMA, accessed on 10 March 2022) in Spanish and (eng, https://forms.gle/mzKhzH8UG8w5mF3 h8, accessed on 10 March 2022) in English. The patients filled out the form daily, recording in each survey the data relative to the previous day. In addition, it was indicated that the form must be completed for each day the patient participated in the study and the patient was reminded that it was essential that the information was as accurate as possible. It was also stated that in the case in which the patient forgot to write down a value and did not remember it, it would be preferable to tag as *I don't remember* this slot.

The survey asked about the patients' feelings regarding mood and stress. For this, the day was divided into six slots of 4-h each starting from 00:00 h, and the patient was asked to assess each slot in relation to mood (sad, normal or happy) and stress (calm, normal or nervous). In addition, data were collected on the quality and duration of nighttime sleep and any naps. The current form includes other options, such as anger, incorporated at the suggestion of the patients. It also includes a section where the patient can specify any other event that she/he considers relevant or unusual in her/his day to day; the usage of drugs was not reported by patients.

Table 1 shows the number of 4-h slots of data per patient for each of the categories of the variables of the survey; stress and mood. The number of days with valid data per patient is also displayed. Not every day has the same number of slots, nor do all the slots have the same number of glucose values. Most days (96.6%) have all the slots completed (six slots per day) with all glucose values (97.4%) filled (16 values per slot).

When collecting data with CGM sensors for long periods of time, it is common to find some data that have not been recorded. To solve this problem, we performed an attribution of the values using interpolation (splines) of degree 3 [11], where the maximum number of consecutive values to be filled is four (maximum correction of 1 h). Patients were asked to complete the information for at least 14 days which corresponds with the expected duration of one Free Style Libre sensor. Patients P1 and P6 continued completing the survey over 50 days so we decided to include all the data in this study. Patients P8 and P10 did not complete the information for all the days. The sensor of patient 5 stopped working on the

fourth day. The CGM data of patient P2 only include valid data of 13 days and patients P3 and P9 include a total of 16 valid days.

Table 2 shows the clinical data of each patient. Sixty percent of the patients were women. The mean HbA1c value is $7.0 \pm 0.8\%$, with a mean weight of $62.5 \pm 10.2$ [Kg], a mean age of $29.1 \pm 10.9$ [years], a mean height of $163.7 \pm 11.2$ [cm] and a mean DX time of $16.5 \pm 9.0$ [years]. Half of the patients were undergoing insulin treatment using Multiple Dose of Insulin (MDI), and the other half via Continuous Subcutaneous Insulin Infusion (CSII).

The inclusion criteria for the patients in the study were as follows:

- Patients with T1DM of at least 1 year since diagnosis (DX);
- Absence of diagnosis of major psychiatric disorder;
- No serious breakthrough disease in the last 6 months.

The study was approved by the Ethics Committee of the Hospital Príncipe de Asturias and all patients signed a prior informed consent form.

**Table 1.** Number of days with valid measurements and number of 4-h slots valid for each category of stress and mood.

| Patient | P1 | P2 | P3 | P4 | P5 | P6 | P7 | P8 | P9 | P10 | Total |
|---|---|---|---|---|---|---|---|---|---|---|---|
| Days | 50 | 13 | 16 | 11 | 4 | 50 | 14 | 8 | 16 | 10 | 192 |
| 4-h slots with valid data and stress survey per patient | | | | | | | | | | | |
| Calm | 0 | 0 | 1 | 0 | 8 | 0 | 0 | 0 | 5 | 0 | 14 |
| Normal | 287 | 55 | 68 | 51 | 7 | 44 | 61 | 39 | 74 | 46 | 732 |
| Nervous | 13 | 21 | 20 | 15 | 3 | 26 | 21 | 4 | 17 | 11 | 151 |
| 4-h slots with valid data and mood survey per patient | | | | | | | | | | | |
| Sad | 0 | 6 | 5 | 5 | 0 | 1 | 4 | 0 | 11 | 4 | 36 |
| Normal | 266 | 72 | 62 | 61 | 16 | 36 | 77 | 30 | 79 | 46 | 745 |
| Happy | 34 | 0 | 22 | 0 | 2 | 0 | 1 | 13 | 6 | 7 | 85 |

**Table 2.** Clinical characterization of each patient. Insulin treatment is performed by Multiple Dose of Insulin (MDI) or by Continuous Subcutaneous Insulin Infusion (CSII).

| Measure | P1 | P2 | P3 | P4 | P5 | P6 | P7 | P8 | P9 | P10 |
|---|---|---|---|---|---|---|---|---|---|---|
| Gender | Woman | Man | Man | Woman | Woman | Woman | Woman | Man | Man | Woman |
| HbA1c% | 6.9 | 7.5 | 7.0 | 6.5 | 7.6 | 5.4 | 8.3 | 7.1 | 6.7 | 7.2 |
| Age [years] | 10 | 25 | 43 | 45 | 27 | 26 | 33 | 18 | 26 | 38 |
| DX time [years] | 2 | 14 | 12 | 35 | 16 | 20 | 16 | 8 | 24 | 18 |
| Weight [kg] | 39.0 | 60.0 | 63.0 | 70.0 | 58.6 | 67.0 | 57.2 | 69.7 | 77.0 | 63.4 |
| Height [cm] | 140 | 159 | 165 | 169 | 159 | 164 | 167 | 168 | 185 | 161 |
| CSII/MDI | MDI | MDI | MDI | CSII | CSII | CSII | CSII | MDI | MDI | CSII |

*2.2. Methodology*

In addition to the HbA1c level, there are different glycemic control measures such as the arithmetic mean or the percentage of time below/above the target range [12], among others, and numerous measures that have been developed to assess glycemic variability. The following measures have been used in this study as glycemic control parameters, as described in the recommendations of the American Diabetes Association [13]:

- Average blood glucose (MEAN). It should be taken into account that FreeStyle Libre sensors present errors, which are usually higher for higher values of glucose [14];
- Percentage of time in hypoglycemia level 1 (hp1) (glucose level < 70 [mg/dL]) and level 2 (hp2) (glucose level $\in$ [54, 70) [mg/dL] );
- Percentage of time in range (Range) (glucose level $\in$ [70, 180] [mg/dL]);

- Percentage of time in hyperglycemia of level 1 (HP1) (glucose level ∈ (180, 250] [mg/dL]) and level 2 (HP2) (glucose level > 250 [mg/dL]).

  As variability parameters, the following were analyzed:

- Mean standard deviation (SD) evaluated every 4 h (SD4h) and 24 h (SD24h);
- Percentage variation coefficient (CV) evaluated every 4 h (CV4h) and 24 h (CV24h);
- Average amplitude of the glycemic excursions (MAGE) evaluated every 4 h (MAGE4h) and 24 h (MAGE24h);
- Average of daily differences (MODD).

The mood and stress perceived by the patients were correlated with blood glucose levels and glycemic variability. To do this, a hypothesis test was carried out to quantitatively compare the data groups of the different measures of variability. We applied the Student's *t*-test (parametric test) using a confidence level of 95% ($\alpha = 0.05$).

## 3. Results

Table 3 shows the different measures of glycemic variability and glycemic averages along with their standard deviations for the 10 patients (P1 to P10). Figure 1 shows the interquartile ranges of glucose values along with their mean values, and the time in the range of each patient.

MEAN, SD, hp1, hp2, Range, HP1 and HP2 measurements were calculated using all glucose values. The SD24h, MAGE24h and CV24h measurements were calculated using the mean glucose values in 24-h slots (one full day). The MODD, SD4h, MAGE4h and CV4h measurements were calculated using the mean of glucose values in 4-h slots.

To correlate the stress and mood with blood glucose levels, the combined categories of the two variables were taken into account. Tables 4 and 5 show the results obtained for the stress and Tables 6 and 7 for mood. The tables show the averages of the different measures of glycemic variability and glycemic averages for the different categories of the variables, and the *p*-values obtained by hypothesis testing for the combined categories.

Here follows a study of the results obtained for the stress. Statistically significant differences have been found (*p*-value < 0.05):

- In the MEAN measure, where the mean increases when the stress increases (calm-normal 126.56 vs. 147.96 [mg/dL], calm-nervous 126.56 vs. 146.13 [mg/dL]);
- In the SD measure, where the variability (in all categories) increases when the stress increases;
- In the CV measure, where the variability (in all categories) increases when the stress increases;
- In the MAGE measure, where the variability (in all categories) increases when the stress increases.

No statistically significant differences (*p*-value < 0.05) were found in the MODD measure.

Here follows a study of the results obtained for mood. No statistically significant differences were found (*p*-value < 0.05):

- In the MEAN measure, however, the patient's mean glucose (in all categories) decreases as mood improves;
- Applying MODD.

  Statistically significant differences were found (*p*-value < 0.05):

- In the SD measure, where the variability increases when the mood changes from sad to happy;
- In the CV measure, where the variability increases when the mood changes from sad to happy;
- In the MAGE measure, where the variability (in all categories) increases when mood improves.

**Table 3.** Averages of the different measures of glycemic variability and glycemic averages together with their standard deviations for the 10 patients (P1 to P10).

| Measure | P1 | P2 | P3 | P4 | P5 | P6 | P7 | P8 | P9 | P10 |
|---|---|---|---|---|---|---|---|---|---|---|
| MEAN [mg/dL] | 145.46 | 158.58 | 142.00 | 199.46 | 139.66 | 119.69 | 143.84 | 156.14 | 136.53 | 145.07 |
| SD [mg/dL] | 53.41 | 69.13 | 54.75 | 61.90 | 63.76 | 45.39 | 70.67 | 58.26 | 56.34 | 58.21 |
| hp1 [%] | 3.51 | 5.37 | 4.05 | 0.57 | 3.86 | 10.22 | 6.42 | 3.44 | 6.31 | 6.40 |
| hp2 [%] | 0.61 | 0.96 | 1.85 | 0.00 | 16.14 | 4.17 | 11.07 | 0.00 | 3.02 | 0.99 |
| Rango [%] | 69.97 | 60.66 | 69.87 | 37.29 | 51.58 | 75.16 | 51.39 | 62.93 | 72.67 | 65.05 |
| HP1 [%] | 25.91 | 33.01 | 24.24 | 62.14 | 28.42 | 10.45 | 31.11 | 33.63 | 18.00 | 27.56 |
| HP2 [%] | 3.95 | 11.62 | 2.99 | 17.88 | 2.81 | 0.48 | 7.20 | 6.58 | 3.28 | 3.42 |
| MODD [mg/dL] | 56.87 ± 29.73 | 78.47 ± 48.15 | 62.55 ± 33.09 | 64.75 ± 40.77 | 30.31 ± 12.91 | 48.65 ± 24.96 | 63.50 ± 34.29 | 60.63 ± 27.23 | 56.49 ± 24.04 | 69.31 ± 25.02 |
| SD4h [mg/dL] | 30.14 ± 15.88 | 33.25 ± 16.67 | 25.36 ± 15.00 | 27.23 ± 13.57 | 24.68 ± 18.24 | 25.40 ± 13.40 | 35.11 ± 22.66 | 38.77 ± 16.08 | 27.36 ± 16.98 | 31.44 ± 17.56 |
| SD24h [mg/dL] | 47.92 ± 11.43 | 54.73 ± 16.45 | 49.28 ± 11.39 | 53.10 ± 7.85 | 41.16 ± 24.57 | 40.90 ± 9.17 | 61.45 ± 16.34 | 54.86 ± 9.75 | 46.78 ± 15.63 | 51.52 ± 14.17 |
| MAGE4h [mg/dL] | 19.60 ± 10.51 | 18.57 ± 9.34 | 11.70 ± 7.32 | 13.71 ± 7.46 | 11.32 ± 8.33 | 14.82 ± 8.78 | 13.10 ± 10.05 | 19.93 ± 13.83 | 11.70 ± 7.29 | 19.09 ± 10.10 |
| MAGE24h [mg/dL] | 31.51 ± 7.75 | 28.60 ± 10.66 | 21.74 ± 9.17 | 27.16 ± 7.94 | 12.19 ± 10.48 | 26.03 ± 8.75 | 21.31 ± 10.44 | 37.33 ± 6.85 | 23.12 ± 6.89 | 31.28 ± 8.05 |
| CV4h [%] | 21.50 ± 10.51 | 22.51 ± 11.60 | 18.80 ± 10.29 | 14.44 ± 7.99 | 18.63 ± 14.20 | 22.10 ± 10.96 | 26.66 ± 15.57 | 25.72 ± 11.04 | 21.09 ± 12.31 | 22.81 ± 11.42 |
| CV24h [%] | 33.04 ± 6.80 | 35.12 ± 8.48 | 34.75 ± 6.55 | 27.06 ± 4.63 | 27.59 ± 13.10 | 34.69 ± 8.55 | 42.95 ± 7.05 | 35.06 ± 4.76 | 34.64 ± 9.67 | 35.97 ± 8.79 |

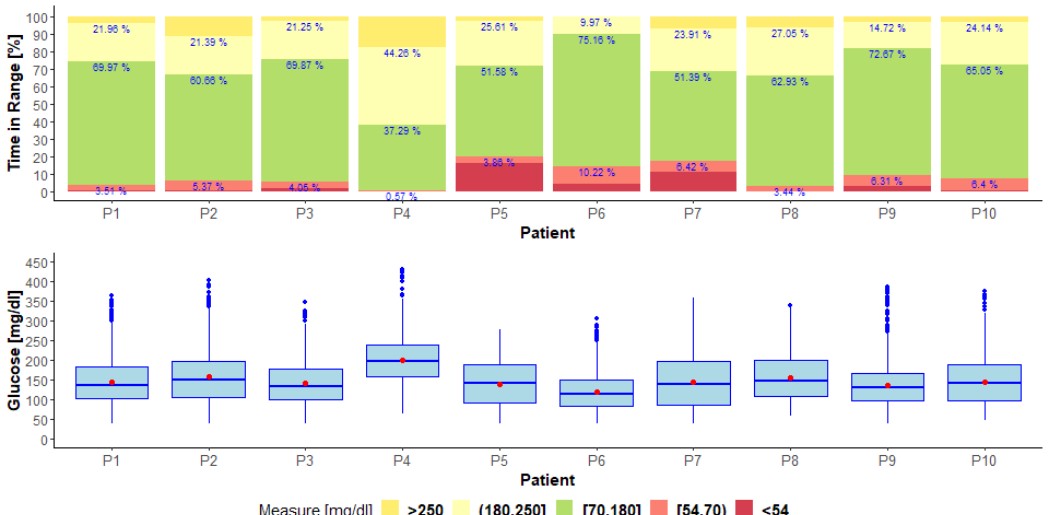

**Figure 1.** The figure above shows the percentage of time the patient has a very low glucose level (<54 [mg/dL]), low ([54, 70) [mg/dL]), in range ([70, 180] [mg/dL]), high ((180, 250] [mg/dL]), and very high (>250 [mg/dL]). The figure below shows the interquartile ranges of glucose. The mean value is represented with a red dot.

**Table 4.** Averages of the different measures of glycemic variability and glycemic averages for the different categories of the stress variable.

| Average | Calm | Normal | Nervous |
|---|---|---|---|
| MEAN | 126.56 | 147.96 | 146.13 |
| SD | 23.55 | 29.31 | 32.95 |
| CV | 20.82 | 20.94 | 24.39 |
| MAGE | 10.97 | 16.27 | 17.27 |
| MODD | 66.59 | 60.19 | 61.67 |

**Table 5.** *p*-values obtained by hypothesis testing for the different measures of glycemic variability and glycemic averages for the different categories of the stress variable.

| Measure | Calm-Normal | Calm-Nervous | Normal-Nervous |
|---|---|---|---|
| MEAN | <0.05 | <0.05 | 0.12 |
| SD | <0.05 | <0.05 | <0.05 |
| CV | 0.89 | <0.05 | <0.05 |
| MAGE | <0.05 | <0.05 | <0.05 |
| MODD | 0.43 | 0.69 | 0.69 |

**Table 6.** Averages of the different measures of glycemic variability and glycemic averages for the different categories of the mood variable.

| Average | Sad | Normal | Happy |
|---|---|---|---|
| MEAN | 151.88 | 148.00 | 146.52 |
| SD | 34.39 | 29.04 | 36.75 |
| CV | 24.53 | 20.99 | 25.14 |
| MAGE | 15.43 | 16.03 | 20.41 |
| MODD | 58.75 | 61.05 | 59.95 |

**Table 7.** *p*-values obtained by hypothesis testing for the different measures of glycemic variability and glycemic averages for the different categories of the mood variable.

| Measure | Sad-Normal | Sad-Happy | Normal-Happy |
|---------|-----------|-----------|--------------|
| MEAN | 0.15 | 0.06 | 0.18 |
| SD | <0.05 | <0.05 | <0.05 |
| CV | <0.05 | 0.33 | <0.05 |
| MAGE | 0.10 | <0.05 | <0.05 |
| MODD | 0.79 | 0.89 | 0.77 |

An important issue that should be considered in future work is the noise and errors associated with CGM systems. In [15] we studied the relationship between the mean value of the instantaneous error with the patients' glycemic variability. The study concludes that there is a relevant correlation between the interstitial glucose error and glycemic variability so that *"the higher the patients' glycemic variability, the higher the instantaneous error mean and dispersion. In other words, the measurements of the sensor have a higher error for those poorly controlled patients with greater glycemic variability"*.

## 4. Conclusions

Variations in mood and stress significantly influence blood glucose levels in patients with T1DM according to the measurements analyzed in this study. Stress and mood affect the blood glucose levels of patients and are, therefore, factors to consider for improving glycemic control.

In general, stress increases the patient's mean and glycemic variability. This trend has been demonstrated in all the categories studied. The mood results have not been as conclusive. The glycemic mean decreases as the patient's mood improves. However, glycemic variability, in general, increases when there is a change in mood, whether the patient is sad or happy, although there are discordant results regarding the different measures of variability.

The MODD measure is not shown as a good indicator of variability since it is the only measure that has not found significant differences in any compared group.

The conclusion of this work should be considered in accordance with the number of participants (10). in addition, very few data for a "calm" state were recorded, which is also a limitation. In future works we will try to eliminate part of the subjectivity in how the information about stress and mood is collected. It would be convenient to incorporate information about stress from levels of hormones measured with electronic devices. The personal feelings recorded by the patients should be definitively enriched, although we consider it valid as a first approximation.

We are conducting a study with 30 patients, collecting information using smartwatches for measuring sleep features, heart rate, calories and steps. Part of this data could be correlated and analyzed to evaluate the level of stress. The work presented in this paper is an initial study and in the future more research methods are needed to evaluate the mood and stress of patients based on biomedical or physiological metrics.

**Author Contributions:** Conceptualization, J.M.V., M.B.-S., E.M. and J.I.H.; Data curation, A.S.-S., A.A. and R.M.; Formal analysis, O.G.; Funding acquisition, O.G., J.L. and J.I.H.; Investigation, J.I.H.; Methodology, J.M.V. and J.I.H.; Software, S.C.; Supervision, M.B.-S. and E.M.; Validation, S.C.; Visualization, S.C.; Writing—original draft, J.I.H.; Writing—review & editing, J.M.V., M.B.-S., E.M. and O.G. All authors have read and agreed to the published version of the manuscript.

**Funding:** This research was funded by Fundación Eugenio Rodríguez Pascual 2019–2020, GLENO Project. Ministerio de Economía y Competitividad under grant TIN2014-54806-R. Ministerio de Ciencia, Innovación y Universidades under grant RTI2018-095180-B-I00. Comunidad de Madrid under grants B2017/BMD3773 (GenObIA-CM) and Y2018/NMT-4668 (Micro-Stress-MAP-CM). European Union through structural and FEDER Funds.

**Institutional Review Board Statement:** The study was conducted according to the guidelines of the Declaration of Helsinki, and approved by the Institutional Review Board (or Ethics Committee) of Hospital Universitario Príncipe de Asturias de Alacalá de Henares, Madrid, Spain (protocol code EC-11/2018, approved 26 November 2018).

**Informed Consent Statement:** Informed consent was obtained from all subjects involved in the study.

**Data Availability Statement:** Data is available under request to absys@ucm.es after signing and collaboration agreement.

**Conflicts of Interest:** The authors declare no conflict of interest.

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
