# Peer review of "Evaluating the Influence of Mood and Stress on Glycemic Variability in People with T1DM Using Glucose Monitoring Sensors and Pools"

_diabetology, doi:10.3390/diabetology3020018_

Round 1

Reviewer 1 Report

The authors have studied whether mood and stress could affect the blood glucose levels of 10 patients with type 1 diabetes. Patients had to report through google forms about their mood and how stressed or not they were. Blood glucose values were recorded every 15 minutes.

  • In the material and methods section, it is not mentioned for how long the patients were followed.
  • The number of days with valid measurements is not consistent at all. It ranges from 4 days to 50 days. ~80% of the patients had less than 15 days with valid measurements.
  • The number of patients is too low and the number of days with valid measurements is too inconsistent to have a solid conclusion.

Author Response

  • In the material and methods section, it is not mentioned for how long the patients were followed.

We added information in the manuscript: "Patients were asked to complete the information for at least 14 days which corresponds with the expected duration of one Free Style Libre sensor. 

  • The number of days with valid measurements is not consistent at all. It ranges from 4 days to 50 days. ~80% of the patients had less than 15 days with valid measurements.

We added the following information in the paper: Patients P1 and P6 continued completing the survey during 50 days so we decided to include all the data in this study. Patients P8 and P10 do not complete the information all the days. The sensor of patient 5 stopped working on the fourth day. CGM data of patient P2 only include valid data of 13 days and contrarily patients P3 and P9 include a total of 16 valid days

  • The number of patients is too low and the number of days with valid measurements is too inconsistent to have a solid conclusion.

We agree with the reviewer, although we consider the study of interest for the readers, in order to put in context our conclusions, we added the following disclaimer text:

Conclusion of this work should be carefully considered. Due to the the different  compliance among patients, it has several limitations. The first one is the number of patients (10), in addition, according to Table 1, very few data for “calm” state were recorded, which again poses a limitation.

Reviewer 2 Report

Questions and comments to the authors:

  1. Please make a clear statement on eventual concomitant usage of drugs that could be confounders such as beta blockers, antidepressants, tranquilizers…
  2. Please explain the wide span of overall days in the study for your patients ranging from 4 to 50, with 3 of 10 patients with less or equal to 10 days. For a relatively small patient size in a multicenter study, with a relatively high percentage of completed slots and glycemic value acquisitions, one would expect more equalized figures tending to the higher number of days in the study. I wonder what was the reason for that: patient compliance, study protocol break, or something else?

Mistyping suggestions:

At the line 91: missing < in front numbers, and misuse of decimal comma instead of decimal stop mark: < 54.70 mg/dl, instead of ([54,70) [mg/dl]).

Also, I guess that all measures (concentrations) and % symbol, currently placed in parentheses [], are presented in this format for some typesetting reason, and that will finally appear without parentheses.

Author Response

  1. Please make a clear statement on eventual concomitant usage of drugs that could be confounders such as beta blockers, antidepressants, tranquilizers…

We added the following text: It (the survey) also includes a  section where the patient can specify any other event that she/he consider relevant or unusual in her/his day to day, eventual usage of drug was not reported by patients

  1. Please explain the wide span of overall days in the study for your patients ranging from 4 to 50, with 3 of 10 patients with less or equal to 10 days. For a relatively small patient size in a multicenter study, with a relatively high percentage of completed slots and glycemic value acquisitions, one would expect more equalized figures tending to the higher number of days in the study. I wonder what was the reason for that: patient compliance, study protocol break, or something else?

We thank the reviewer for this important comment, we added the following information accordingly: "Patients were asked to complete the information for at least 14 days which corresponds with the expected duration of one Free Style Libre sensor. Patients P1 and P6 continued completing the survey during 50 days so we decided to include all the data in this study. Patients P8 and P10 do not complete the information all the days. The sensor of patient 5 stopped working on the fourth day. CGM data of patient P2 only include valid data of 13 days and contrarily patients P3 and P9 include a total of 16 valid days."

Mistyping suggestions:

At the line 91: missing < in front numbers, and misuse of decimal comma instead of decimal stop mark: < 54.70 mg/dl, instead of ([54,70) [mg/dl]).

In reality the text include intervals so, we correct the text accordingly

Also, I guess that all measures (concentrations) and % symbol, currently placed in parentheses [], are presented in this format for some typesetting reason, and that will finally appear without parentheses

corrected

Reviewer 3 Report

The manuscript entitled “Evaluating the influence of mood and stress on glycemic variability in people with T1DM using glucose monitoring sensors and pools” proposes an experiment to assess the influence of mood and stress in glycemic variation for people with T1DM. The methods and results are well described. However, the scientific discussion is poor, it should be enhanced. Before further consideration, the following points should be addressed:

1. A few more keywords should be added. For example: Continuous Glucose Monitoring or glycemic control.
2. The selection of FreeStyle Libre sensors from Abbott should be briefly justified.
3. An English version of the survey to collect information on mood and stress should be provided, maybe as a supplementary material.
4. The limitations of the study should be clearly discussed. For example, the mean error of FreeStyle Libre sensors should be discussed, and how this error can impact the reported results. The number of patients (10) is a limitation as well, and it should be discussed. According to Table 1, very few data for “calm” state were recorded, which again poses a limitation. Also, the subjective way in which the information about stress and mood is collected, based on personal feelings by the patients, should be discussed. As an initial study, this seems an acceptable method, but its influence in the results should be discussed, and some more accurate methods to evaluate the patient’s state based on biomedical or physiological metrics should be discussed for future studies.
5. The conclusions might need to be moderated according to the above-mentioned discussions.

Author Response

1. A few more keywords should be added. For example: Continuous Glucose Monitoring or glycemic control.

Added

2. The selection of FreeStyle Libre sensors from Abbott should be briefly justified.

We added the following explanation: FreeStyle Libre sensors were selected because for some of the patients was covered by the Spanish National Health System and they already know the particularities of those sensors

3. An English version of the survey to collect information on mood and stress should be provided, maybe as a supplementary material.

A new english version was included.

4. The limitations of the study should be clearly discussed. For example, the mean error of FreeStyle Libre sensors should be discussed, and how this error can impact the reported results. The number of patients (10) is a limitation as well, and it should be discussed. According to Table 1, very few data for “calm” state were recorded, which again poses a limitation. Also, the subjective way in which the information about stress and mood is collected, based on personal feelings by the patients, should be discussed. As an initial study, this seems an acceptable method, but its influence in the results should be discussed, and some more accurate methods to evaluate the patient’s state based on biomedical or physiological metrics should be discussed for future studies.
5. The conclusions might need to be moderated according to the above-mentioned discussions.

We thank te reviewer for this comment. Accordingly we added the following text in the paper:

"t should be taken into account that FreeStyle Libre sensors present errors, which is usually higher for higher values of glucose [14]"

"An important issue that should be considered in future work is the noise and errors associated to CGM systems. In [15] we  studied the relationship between the mean value of the instantaneous error with the patients’ glycemic variability. The study concludes that there is a relevant correlation between the interstitial glucose error and  glycemic variability so that "the higher the patients’ glycemic variability, the higher the instantaneous error mean and dispersion. In other words, the measurements of the sensor have a higher error for those poorly controlled patients with greater glycemic variability"

"Conclusion of this work should be  considered in accordance with the number of participants (10). in addition, very few data for “calm” state were recorded, which it also  a limitation.  In future works we will try to eliminate part of the subjectivity in the information about stress and mood is collected. It would be convenient to incorporate information about stress from levels of hormones measured electronic devices. The personal feelings records by the patients should be definitively enriched, although we consider it valid as a first approximation."

"We are  conducting a study with 30 patients, collecting information using smartwatches for measuring sleep features, heart rate, calories and steps. Part of this data could be correlated and be analysed for evaluating the level of stress. The work presented in this paper is initial study and in the future  more research methods are needed to evaluate mood and stress of patients based on biomedical or physiological metrics."

Round 2

Reviewer 1 Report

Concerns were addressed 

Reviewer 3 Report

All my prior comments have been satisfactorily addressed, and the associated modifications of the manuscript have increased its quality.